# Rapid elongation drives the exceptionally fast aggregation of the most common localized human amyloid medin

Vaidehi Roy Chowdhury [1], Robert I. Horne [1], Mariana P. Cali[1], Zenon Toprakcioglu[1], Sara Linse [2] & Michele Vendruscolo [1] ✉

Amyloid deposition is a hallmark of numerous age-related diseases, and understanding the chemical mechanisms that govern amyloid formation is crucial for advancing the rational development of protein aggregation inhibitors. With amyloid formation rates varying widely across proteins, here we report the quantitative aggregation mechanism of medin, the most common localized amyloid in humans, and find it to be much faster compared to well-known pathological amyloids such as amyloid-β (Aβ), tau and α-synuclein. We report the microscopic rate constants and reaction orders of medin fibril formation by monitoring the aggregation of recombinant human medin in vitro via a fluorescence-based assay, global kinetic modeling, secondary structure analysis and electron microscopy. Medin spontaneously forms amyloid fibrils at physiological pH and temperature in quiescent solution at concentrations as low as 25 nM, with the highest fibril elongation rate constant when compared to those of Aβ, tau and α-synuclein. Our results identify the microscopic basis of the widespread observation of medin aggregates upon aging, offering a mechanistic starting point for drug discovery to inhibit medin aggregation.

Amyloid fibril formation is a crucial process of great functional[1–4] and pathological relevance[5–7]. Elucidating the mechanism of pathological amyloid formation lies at the heart of dissecting the origins of over 50 human diseases[5–7] and developing therapeutics to combat aggregation-induced pathogenesis[8–10]. While considerable progress has been made in understanding the microscopic mechanisms of aggregation of well-known pathological amyloids such as amyloid-β (Aβ)[11,12], tau[13–15], α-synuclein (αSyn)[16–18], prion[19,20], and amylin or islet amyloid polypeptide (IAPP)[21,22], many of the remaining 37 known disease-related amyloids remain relatively unexplored[5–7,23].

In this work, we focus on medin, a protein fragment that forms the most common senile localized amyloid in humans, called aortic medial amyloid, found in almost every European above the age of 60[24]. It deposits with age in the wall of the aorta, the main and the largest artery supplying oxygenated blood directly from the heart to the rest of the human body[25,26]. Medin is a 50-residue peptide formed by post-translational cleavage of the glycoprotein lactadherin[24,27], which is important for apoptotic body removal[28,29] as well as Aβ plaque clearance in Alzheimer's disease (AD)[30–33]. Medin aggregation is implicated in multiple age-related vascular diseases of the upper body[34] (Fig. 1), including giant cell arteritis (the most common arterial inflammatory disease)[35], thoracic aortic aneurysm[36,37], and aortic

dissection[37]. Medin cross-seeds serum amyloid A aggregation in systemic amyloid A (AA) amyloidosis[38]. More recently, medin has been shown to coaggregate with and enhance the deposition of cerebrovascular Aβ in both human patients and AD mouse model, causing cerebral amyloid angiopathy (CAA) in AD, while medin deficiency in animal models decreased Aβ deposition by half[39–41]. Medin aggregates are increased in cerebral arterioles of patients with vascular dementia or AD compared to cognitively healthy controls[41]. Among cerebrovascular pathologies, arteriolar medin was found to be the best predictor of AD diagnosis independent of plaque load or tau burden[40,41]. A splice acceptor variant of *MFGE8* (the gene encoding lactadherin) lacking residues 268–319 that harbor two of the three key amyloidogenic regions of medin[42,43] provides significant protection against coronary atherosclerosis and myocardial infarction in Japanese and several European ethnicities[44]. Despite extensive disease association, the detailed mechanism of medin aggregation has remained poorly understood over the past 26 years since its discovery[24], thereby considerably delaying therapeutic endeavors.

To bridge this knowledge gap, we determined the microscopic steps underlying medin aggregation using a chemical kinetics framework[45–47]. We monitored medin aggregation kinetics across a range of monomer and seed concentrations using thioflavin T (ThT), a fluorescent reporter of amyloid formation, and applied global kinetic modeling[46] to extract the microscopic

[1]Centre for Misfolding Diseases, Yusuf Hamied Department of Chemistry, University of Cambridge, Cambridge, UK. [2]Biochemistry and Structural Biology, Department of Chemistry, Lund University, Lund, Sweden. ✉e-mail: mv245@cam.ac.uk

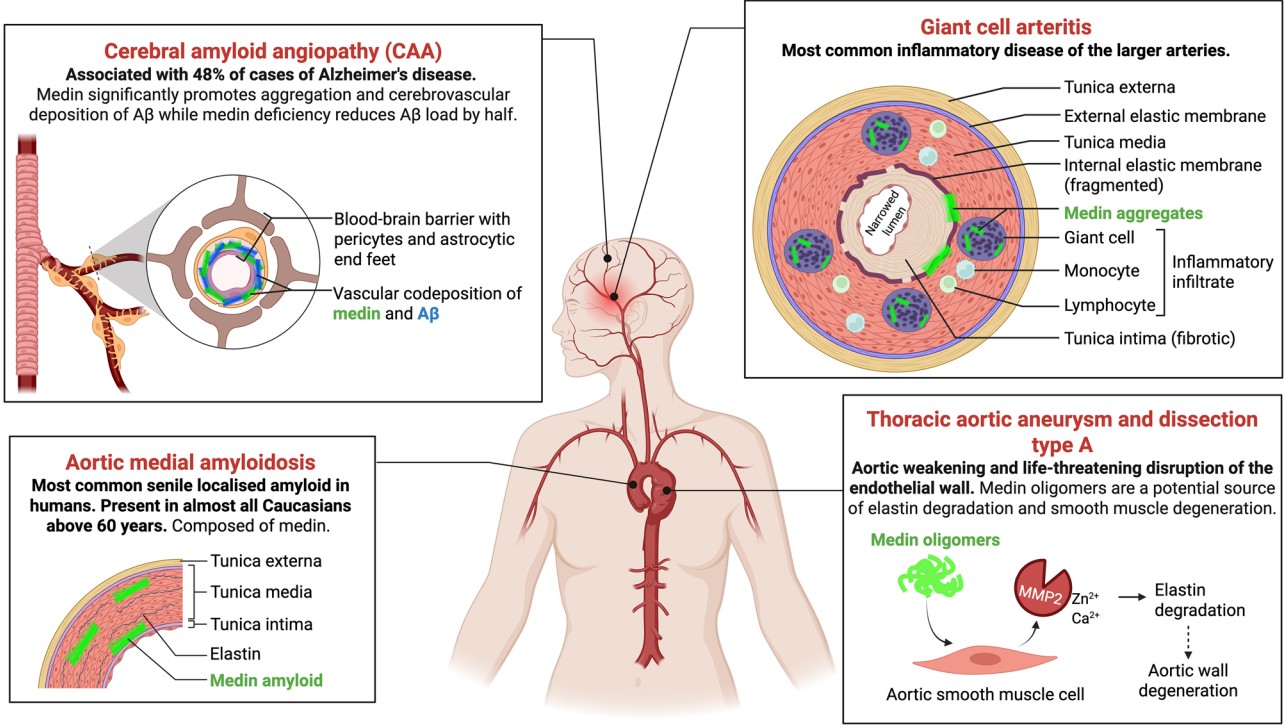

**Fig. 1 | Amyloid and pre-fibrillar aggregates of medin are implicated in the pathogenesis of several vascular diseases in humans.** Medin oligomers are found to be the most toxic aggregation intermediate in vitro and are responsible for the pathogenic consequences[37]. Adapted from refs. 24,35,37,40,41.

rate constants and reaction orders for fibril nucleation, elongation, and fragmentation processes[48–50]. We found that medin aggregates with exceptionally high speed at nanomolar concentration in physiological pH and temperature in vitro. It is driven by rapid primary nucleation (reaction order ~1) and elongation (rate constant ~$10^7\,s^{-1}$) that surpass those of other known pathological amyloids[11–16,51], and sustained by saturated secondary processes that proceed independently of monomer concentration (reaction order ~0).

Our results define an extreme kinetic paradigm in pathological protein self-assembly, characterized by rapid nucleation events with weak monomer dependence coupled to fast elongation. These findings provide a quantitative mechanistic rationale for the widespread accumulation of medin aggregates with age. They also introduce a framework for elucidating the influence of extracellular milieu on medin aggregation and establish a platform for screening aggregation inhibitors.

## Results
### Aggregation of recombinant medin into amyloid fibrils in vitro
Medin was produced recombinantly to >95% purity at a yield of ~3 mg/L of bacterial culture using a protocol[52] (see "Methods") that provides improved peptide integrity and purity compared to previous approaches reported in literature[53,54] (Fig. 2A). The previous procedure for recombinant production of medin yielded a peptide integrity of >70%, as determined by western blotting and peptide sequencing by mass spectrometry[53]. Solid-phase synthesis of medin[54] is not recommended for aggregation kinetic studies as chemical synthesis often leaves behind traces of undesired organic solvents, truncated or incorrectly synthesized peptides, and other byproducts, leading to loss of bioactivity[55–60]. With the recombinant protein production protocol that we developed in this study, no other proteins could be observed by sodium dodecyl sulfate-polyacrylamide gel electrophoresis (SDS-PAGE) (Fig. 2A), the peptide integrity was >95% as measured by intact protein mass spectrometry (Fig. 2B) and >90% as measured by bottom-up proteomics (Supplementary Table 1). Dynamic light scattering (DLS) experiments (Fig. 2C and Supplementary Fig. 1B) showed that monomeric and initial oligomeric species of medin have hydrodynamic diameters in the range of

0.4–6 nm. After quiescent incubation at pH 7.4 and 37 °C, medin forms aggregates that predominantly increase in hydrodynamic diameter with time, possibly owing to fibril maturation, and fall in the range of 200–3200 nm. Using circular dichroism (CD) spectroscopy (Fig. 2D), monomeric medin was found to be predominantly disordered with a highly negative mean residual ellipticity at around 200 nm and a low amplitude (trough) at around 222 nm, corroborating previous reports of its intrinsic disorder[61,62]. Upon overnight incubation at 37 °C and pH 7.4, medin at 10 µM formed amyloid fibrils with a characteristic β-sheet structure, with a positive signal at 195 nm and a trough at 218 nm. A fibrillar morphology was also confirmed by transmission electron microscopy (TEM) (Fig. 2E).

### Saturated secondary pathways sustain medin aggregation in vitro
Amyloid fibril formation is governed broadly by three classes of microscopic processes: (1) nucleation of aggregates, (2) elongation of protofilaments (sometimes preceded by an oligomer conversion step), and (3) fibril maturation[11,63]. Nucleation processes can either be spontaneous, involving pure monomers only (primary pathways), catalyzed by existing heterotypic surfaces (heterogeneous primary nucleation[64,65]) or by catalytically active surfaces or ends of pre-formed fibrils (secondary pathways)[11,14,22,45,66–71]. Secondary pathways can either depend on the monomer concentration (monomer-dependent secondary nucleation[11]) or the rate could be independent of monomer concentration and instead be limited by the availability of filament ends (as in fibril fragmentation[19,71]) or rate of detachment of secondary oligomers from fibril surfaces (saturated secondary nucleation[12]). To determine the relative contributions of each nucleation process in medin amyloidogenesis, we investigated the aggregation process of medin using the fluorescent reporter of amyloid fibril formation, thioflavin T (ThT)[72], and performed global kinetic modeling[46]. ThT binds to medin amyloid fibrils, emitting at 480 nm upon excitation at 440 nm, and increasing its quantum yield in the process. It has a maximum intensity that linearly depends on the initial monomer concentration if the ThT concentration is tailored appropriately (Supplementary Fig. 2). When purely

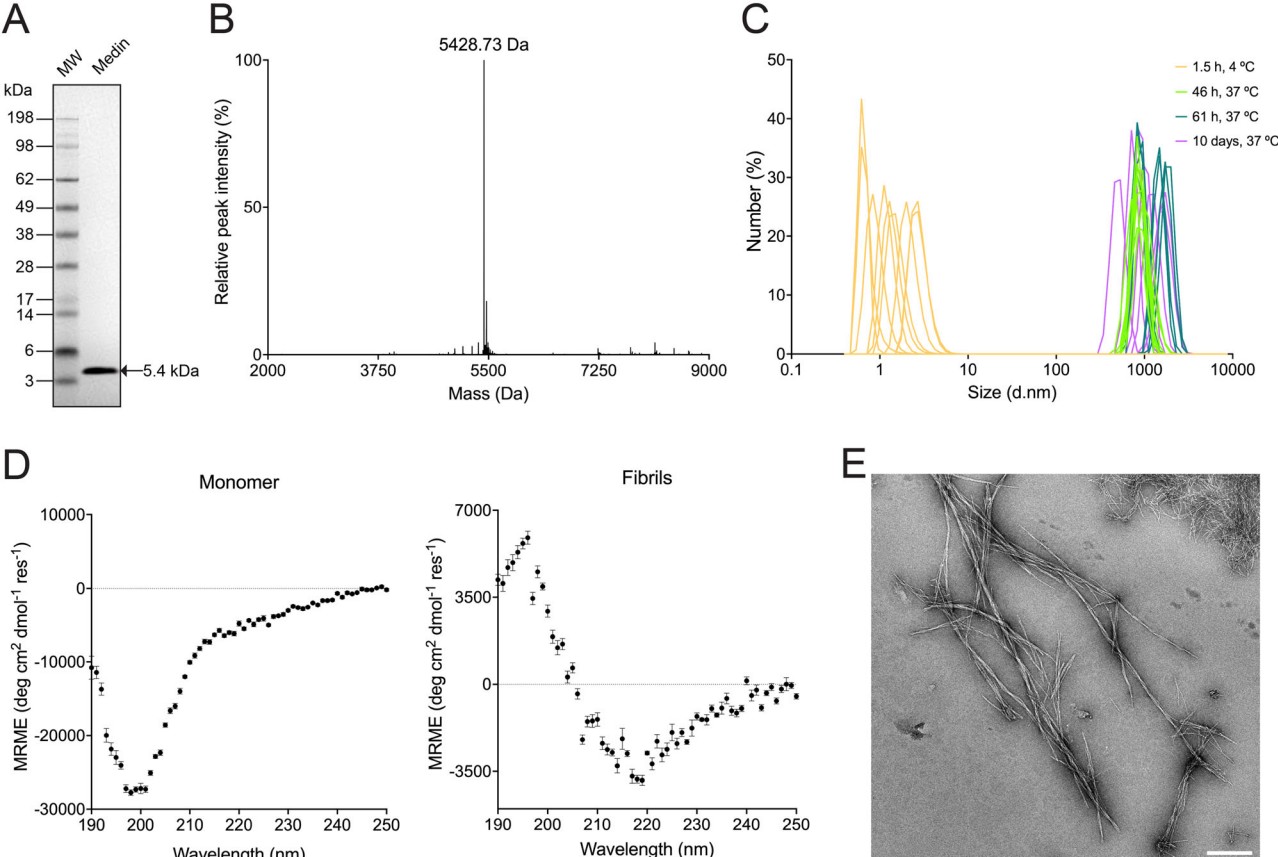

**Fig. 2 | Recombinant medin forms amyloid fibrils in vitro under near-physiological conditions (10 µM medin, pH 7.4, 37 °C). A** Purity measurement by SDS-PAGE followed by Coomassie blue staining using 2.5 µg of recombinant medin. **B** Confirmation of medin integrity by electrospray ionization-mass spectrometry (ESI-MS). Theoretical molecular weight = 5430.98 Da. **C** Monitoring change in particle size due to fibrillation of medin by DLS ($N = 3$, $n = 3$). **D** Far-UV CD spectra of monomeric and fibrillar medin at 10 µM monomer-equivalent concentration. Monomeric medin exhibits a highly negative signal at 200 nm and a low amplitude signal at 222 nm, characteristic of disordered proteins and peptides. The CD spectrum of fibrillar medin exhibits a trough at 218 nm, characteristic of a β-sheet. Fibrils were prepared by quiescent overnight incubation at 37 °C, followed by brief and mild sonication. Data are presented as mean ± SEM. MRME = mean residue molar ellipticity. **E** Transmission electron micrograph of negative-stained fibrils formed by overnight incubation of 10 µM monomeric medin at pH 7.4 at 37 °C under quiescent conditions. Typical amyloid fibrillar morphology was observed. Scale bar = 500 nm.

monomeric medin (Supplementary Fig. 1) is incubated at different concentrations at physiological pH and temperature in the presence of ThT, the ThT fluorescence intensity at 480 nm shows a characteristic sigmoidal increase over time (Fig. 3A). We studied the kinetics of aggregation in the lower range (25–625 nM) of the physiological concentration range of medin (0–13.72 µM in the wet aortic tissue of humans above 55 years of age[26]), where the logarithm of aggregation half-time ($t_{1/2}$) scaled negatively with that of the initial monomer concentration ($m_0$) (Fig. 3B and Supplementary Fig. 3A, B). This defined the kinetic regime for mechanistic calculations. Notably, no aggregation was noticed in 10 nM medin monomer over 16 h, suggesting that the critical monomer concentration for aggregation lies between 10 nM and 25 nM monomer (Supplementary Fig. 3C, D). Our results generated the scaling exponent $\gamma = -0.5$ of the variation of $t_{1/2}$ with $m_0$ (Fig. 3B), which is characteristic of saturated secondary processes (fragmentation or secondary nucleation), i.e., secondary processes that depend weakly on the initial monomer concentration[46]. To further confirm the presence of secondary pathways in medin aggregation in vitro, we performed low-seeded aggregation assays, whereby incubation of medin monomer with pre-formed fibrils (seeds) at 1% seed:monomer ratio was found to noticeably decrease the $t_{lag}$ and $t_{1/2}$ of fibril formation while retaining a sigmoidal profile and similar mechanism of aggregation (Fig. 3C and Supplementary Fig. 4). We note that the quantitative estimates of rate constants and reaction orders are primarily constrained by the well-fitted intermediate concentrations (100–500 nM), whereas the boundary points

(50 nM and 625 nM), which show modest deviations from the global model, do not influence the inferred reaction parameters.

To understand the effect of ionic strength on the solubility and aggregation propensity of medin, amyloid fibril formation was monitored using ThT in the presence of increasing concentrations of NaCl, specifically no salt, hypotonic (40 mM or 0.2%), physiologically normal (150 mM or 0.9%), and hypertonic (500 mM or 3%) solutions (Supplementary Fig. 5A, B). The amplitude of ThT fluorescence intensity is higher in the presence of salt than in the absence thereof (Supplementary Fig. 5C). One possible reason to explain this phenomenon could be that an increase in ionic strength leads to an increased binding of ThT to the amyloid fibrils[73]. However, the aggregation $t_{1/2}$ either decreases or does not change in the presence of salt compared to in the absence thereof (Supplementary Fig. 5D and Supplementary Table 2). Importantly, increasing concentrations of salt do not alter the nucleation mechanisms, as evident in the relatively similar scaling exponents of $\ln t_{1/2}$ to $\ln m_0$ at all salt concentrations (Supplementary Fig. 5E and Supplementary Table 3). Therefore, increasing concentrations of physiologically relevant salts such as NaCl do not appreciably change the solubility and aggregation propensity of medin.

### Determination of microscopic rate constants and reaction orders via kinetic modeling and global data analysis

We next sought to determine the microscopic steps constituting medin aggregate proliferation and growth. We monitored the aggregation

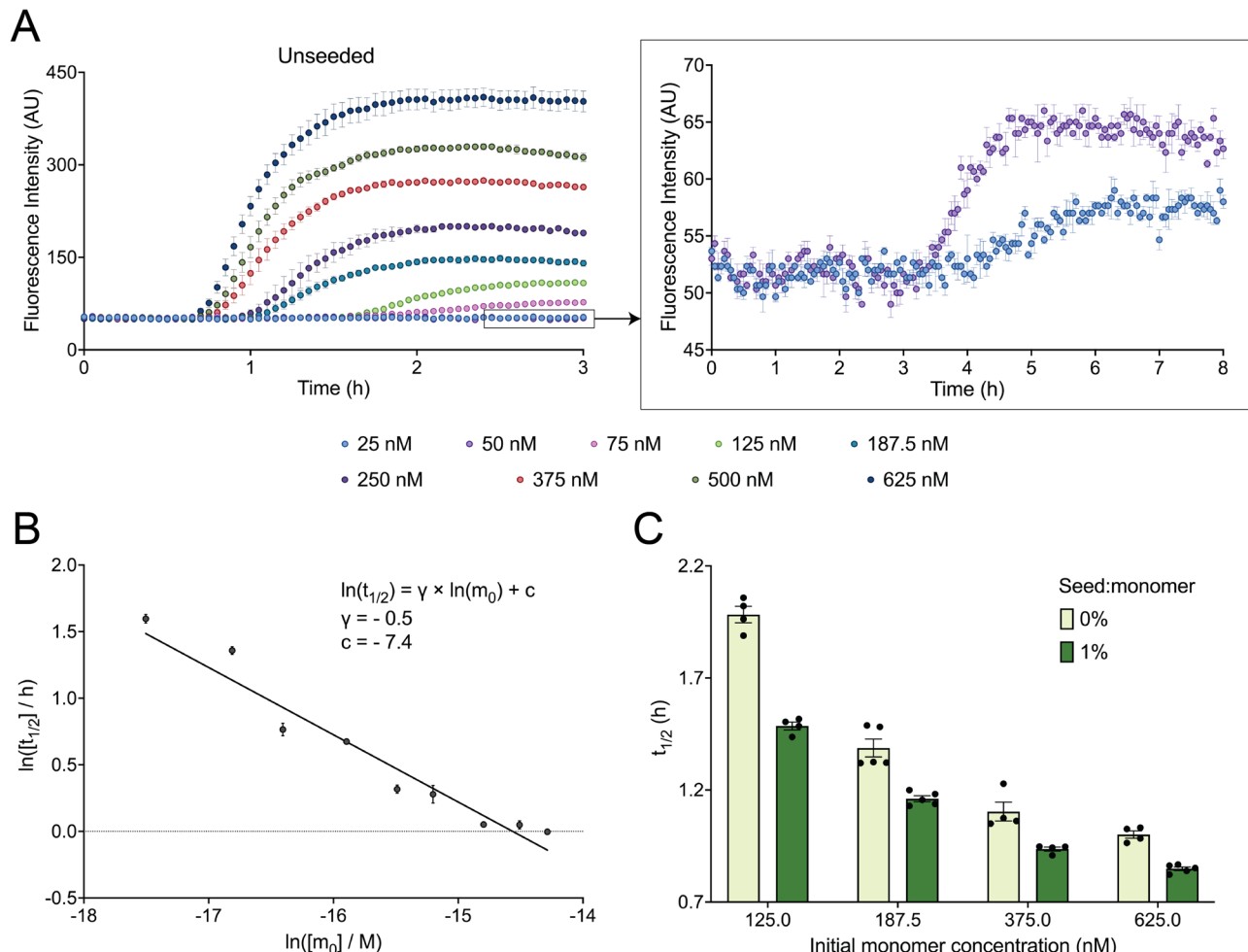

**Fig. 3 | Secondary pathways contribute to medin aggregation under near-physiological conditions in vitro. A** Recombinant medin monomers were incubated at increasing concentrations between 25 and 625 nM (different colors) at pH 7.4 and 37 °C. Aggregation was monitored using ThT. The inset shows the full time-course of aggregation of medin at 25 and 50 nM concentrations over a period of 8 h. **B** The half-times ($t_{1/2}$) of these reactions depend inversely on the square-root of the initial monomer concentrations ($m_0$), which is characteristic of fragmentation or saturated secondary nucleation ($\gamma = -0.5$; $R^2 = 0.93$). **C** Incubation of monomers with pre-formed fibrils (seeds) at a 1% seed:monomer ratio accelerates the rate of aggregation, as shown by the reduced $t_{1/2}$ values of the seeded reactions (dark green) compared to the unseeded reactions (light green) at the same initial monomer concentrations. Data are presented as mean ± SEM; AU arbitrary unit.

kinetics of medin across a range of monomer and seed concentrations and globally fitted the resultant data to a kinetic model of amyloid formation[46] (see Supplementary Note 1). According to this model, aggregation in monomeric solutions begins with homogeneous primary nucleation that forms initial aggregates that can then elongate by monomer addition into fibrillar structures. These fibrils, in turn, can catalyze the formation of new aggregates by secondary processes. The overall aggregate proliferation depends on the combined rate constants $k_+k_n$, $k_+k_2$ and $k_+k_-$ as well as reaction orders $n_c$ and $n_2$. Here, $k_n$, $k_+$, $k_2$ and $k_-$ are the rate constants for primary nucleation, elongation, secondary nucleation, and fragmentation, respectively, while $n_c$ and $n_2$ are reaction orders for primary and secondary nucleation respectively[46].

To calculate the elongation rate constant $k_+$, high-seeded assays were performed by quiescently incubating medin monomers with medin seeds at 20–50% seed:total protein ratio at 37 °C (Fig. 4A and Supplementary Fig. 7). At such high seed:monomer ratios, the reaction is dominated by elongation with negligible nucleation events[46,49], thus enabling the determination of $k_+$, which in the case of medin was found to be $5 \times 10^7\,\mathrm{M}^{-1}\,\mathrm{s}^{-1}$.

Considering the possibilities of either secondary nucleation or fragmentation dominated proliferation of medin aggregates, low-seeded assays

at 1% seed:monomer ratio were performed at different initial monomer concentrations (Fig. 4B). The resulting data were fitted separately by either secondary nucleation or fragmentation-dominated kinetic models, by setting the elongation rate constant to the previously fitted value. According to the kinetic model of aggregation process dominated by fragmentation, the fragmentation rate constant $k_-$ was determined to be $4.3 \times 10^{-8}\,\mathrm{s}^{-1}$. This is of the same order of magnitude as the fragmentation rate constant of murine prion aggregation initiated with pre-formed fibrillar seeds ($k_- \geq 1.6 \times 10^{-8}\,\mathrm{s}^{-1}$)[19]. According to a secondary nucleation dominated model, the secondary nucleation rate constant $k_2$ was determined to be $4.4 \times 10^{-8}\,\mathrm{s}^{-1}$. The reaction order $n_2$ for secondary nucleation was found to be negligibly small (~0), which indicates monomer-independent or saturated secondary nucleation.

By fitting the data from the unseeded assay (Fig. 4C) using the pre-determined kinetic parameters and secondary nucleation- or fragmentation-dominated spontaneous aggregation model, the primary nucleation reaction order ($n_c$) was found to be ~1 (Supplementary Table 4). According to a fragmentation-dominated model of protein aggregation, unseeded reactions of medin had a $k_+ k_n$ value of $3.3 \times 10^{-1}\,\mathrm{M}^{-1}\,\mathrm{s}^{-2}$. A secondary nucleation dominated model, however, predicted a value of $2.3 \times 10^{-2}\,\mathrm{M}^{-1}\,\mathrm{s}^{-2}$ (Supplementary Table 4).

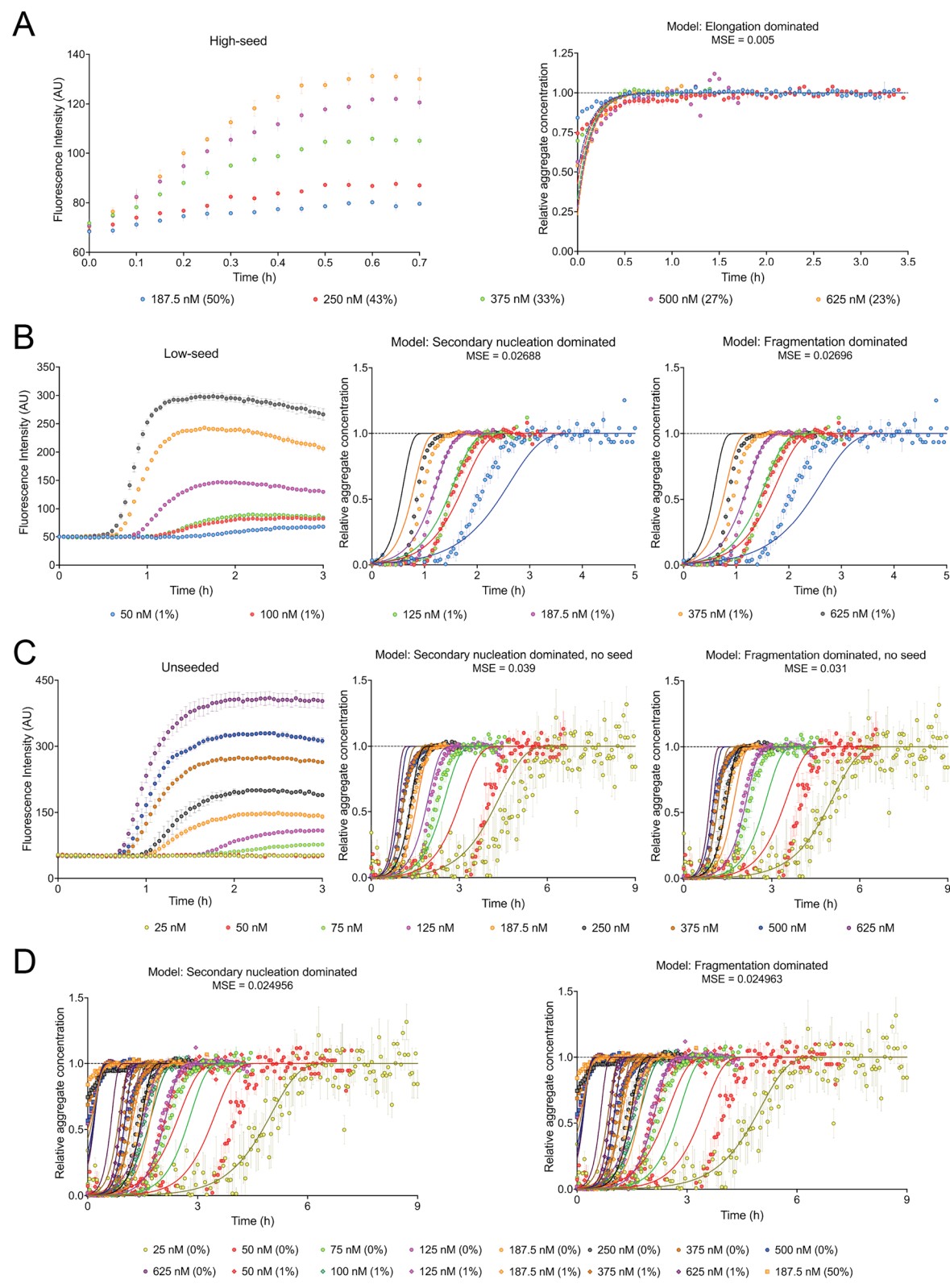

For the determination of the overall primary nucleation rate constant, we performed a global analysis of data from all the assays in the presence or absence of seed (Fig. 4D) and fitted the data separately to either secondary nucleation or fragmentation models. In all the assays, both the models fit the data equally well, as evident from the relatively similar mean squared error (MSE) values. The primary nucleation rate constant was determined to be $k_n = 1 \times 10^{-8}\,\mathrm{s}^{-1}$, and the corresponding reaction order was determined, $n_c = \sim 1$ (Supplementary Table 5). This value suggested a very low nucleation energy barrier. The fitted parameters were further verified by conducting replicate experiments of the unseeded assays using medin produced in different batches, and were found to be consistent across batches (Supplementary Table 6).

**Fig. 4 | Determination of kinetic parameters of microscopic processes involved in recombinant medin aggregation in vitro. A** High-seeded assays performed with medin monomers (187.5–500 nM; different colors) in the presence of high amounts of pre-formed fibrils (187.5 nM) exhibit a characteristic exponential-plateau shape of elongation of existing fibril ends. These assays enabled the determination of the elongation rate constant $k_+$ ($5 \times 10^7 \, \mathrm{M^{-1} \, s^{-1}}$) by fitting the normalized data to negligible rate constants and reaction orders for primary and secondary processes. Seed:total protein molar ratios are mentioned in parentheses. **B** Data from low-seeded assays performed at 1% seed:monomer molar ratio (50–625 nM monomer; different colors) were normalized and fitted separately to secondary nucleation and fragmentation dominated models. The rate constant of the monomer-independent secondary pathway ($k_2$ or $k_-$) was found to be $4 \times 10^{-8} \, \mathrm{s^{-1}}$. **C** Data from unseeded aggregation assays were normalized (different colors denote different initial

monomer concentrations) and separately fitted to secondary nucleation and fragmentation dominated spontaneous aggregation kinetic models to show that the primary nucleation step has a first-order dependence on the initial monomer concentration. **D** Global fitting of aggregation kinetics data obtained under various degrees of seeding in the previous three assays (different colors denote different initial monomer concentrations; circles denote unseeded, diamonds low-seeded, and squares high-seeded reactions) fitted to either a saturated secondary nucleation ($n_2 = 0$) or a fragmentation-dominated model determined the primary nucleation rate constant $k_n$ to be $1 \times 10^{-8} \, \mathrm{s^{-1}}$. Seed:monomer ratios are indicated in parentheses beside initial monomer concentrations (different colors) in the legend. Symbols denote data points and lines denote fits. Data are presented as mean ± SEM; AU arbitrary unit.

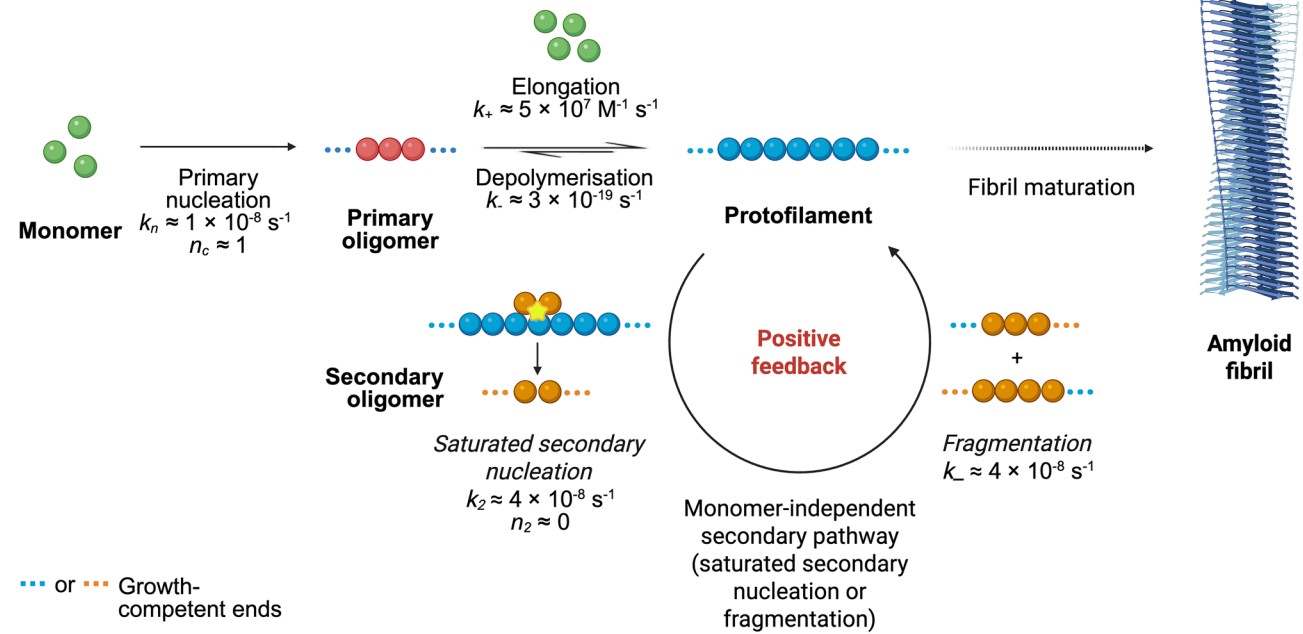

**Fig. 5 | Schematic diagram illustrating a kinetic mechanism of medin aggregation consistent with the in vitro aggregation measurements reported in this work.** The values of the kinetic parameters obtained for each microscopic step are shown. Primary and saturated secondary processes contribute almost equally to the

proliferation of medin aggregates, while exceptionally rapid elongation leads to the fast aggregate growth of medin even at submicromolar concentrations under physiological conditions.

## Discussion

We reported a quantitative determination of the kinetic mechanism of aggregation of medin, the most common senile localized amyloid in humans. Our results indicate that medin aggregation is dominated by an exceptionally rapid fibril elongation step. First-order primary nucleation and saturated secondary processes, which depend weakly on the initial monomer concentration, account for medin aggregate proliferation, contributing to nucleation at rates of the same order of magnitude (Fig. 5). These results quantify a previous hypothesis of nucleation-dependent growth of amyloid fibrils of medin[53] by uncovering the microscopic steps involved in the aggregation process and the corresponding rate parameters. We also reported an improved protocol for consistent production of kinetically conserved batches of recombinant human medin with much higher peptide integrity and purity compared to previous protocols without introducing the possibility of racemization[53,54].

Medin has a higher elongation rate constant than most commonly known amyloids, such as Aβ, α-synuclein, and tau[11–16] (Fig. 6A). For most well-studied pathological amyloids, such high elongation rates are only achieved in the presence of cofactors, such as heparin for tau[74]. The fast aggregation of medin may also be demonstrated by the comparison

that ~900 nM Aβ42 aggregates with the same half-time as that of 125 nM medin under the same environmental conditions (Fig. 6B, C, Supplementary Fig. 8 and Supplementary Note 2). This pinpoints to the elongation step as a potential target for drug discovery to prevent disease-associated medin aggregation, suggesting the development of inhibitors binding either to the monomers and stabilizing them in their native state, or to fibril ends to prevent further monomer addition.

Notably, our results imply a low critical concentration and low nucleation barriers, accounting for the fast aggregate growth even at sub-micromolar concentrations. The critical concentration of medin aggregation is as low as ~14 nM. These results suggest a fast conversion of monomeric and oligomeric medin into amyloid fibrils with a highly transient existence of free oligomeric intermediates of low population. Further oligomer flux measurement experiments may help verify this hypothesis[63]. It may also be relevant to investigate in the future if there exist certain concentration sub-regimes within the overall kinetic regime (25–625 nM) where the relative contribution of each microscopic process towards nucleation of new aggregates may differ[12].

We found that possible on-pathway intermediates in medin aggregation include both primary (products of homogeneous primary

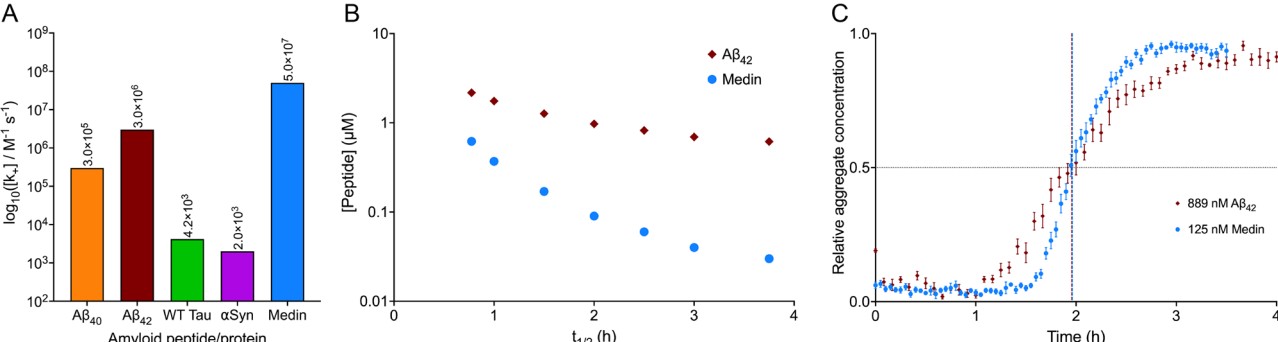

**Fig. 6 | Comparison of the elongation rate constant of medin with other human amyloids. A** Medin (blue) has a higher elongation rate constant ($k_+$) at pH 7.4 and 37 °C compared to the other amyloids, as reported in the literature. Aβ$_{40}$ (orange) was assayed at the same pH and temperature as medin[12]. Aβ$_{42}$ (brown) kinetics were determined in 20 mM sodium phosphate (NaPi) 0.2 mM EDTA 0.02% NaN$_3$ pH 8.0 quiescently at 37 °C[11]. Wild-type (WT) tau (0N4R isoform; green) was aggregated quiescently in saline-sodium phosphate-ethylene diamine tetraacetate (SSPE) buffer pH 7.4 at 37 °C, and the aggregation kinetics were determined by total internal reflection fluorescence (TIRF) microscopy[13]. α-Synuclein (αSyn; purple) aggregation was monitored in phosphate-buffered saline (PBS), pH 7.4, quiescently at 37 °C in the presence of pre-formed fibrils[16]. **B** It requires almost 10- to 100-fold higher concentration of Aβ$_{42}$ (brown diamond) to aggregate spontaneously with the same half-time as that of medin (blue circle). **C** 125 nM medin (blue circle) aggregates with the same $t_{1/2}$(1.9 h) as 889 nM Aβ$_{42}$ (brown diamond) at pH 7.4 and 37 °C.

nucleation) and secondary (products of either secondary nucleation or fragmentation) oligomers. This finding is significant because of the likely importance of medin oligomers in mediating toxicity in humans. Medin has been shown to mediate several vascular diseases of the upper body[26,34–37,40,41]. Recent studies on many amyloidogenic peptides and proteins, including medin[35], suggest that the on-pathway intermediates in solution phase are often the main pathogenic species and not the resultant fibrils[75]. Previously, medin oligomers have also been shown to form pores in lipid bilayers in vitro, suggesting a potential mechanism for destabilizing vascular permeability[76]. The current study reveals the kinetic mechanisms by which these oligomers are formed—both primary and secondary processes are sources of medin oligomers, and they operate at rates that are of the same order of magnitude. Further areas to investigate are the misfolding patterns of primary vs secondary oligomers[63] and whether they exert similar levels of pathological effects on the human tissue. This would, in turn, reveal if it would be more beneficial to therapeutically stabilize the native monomeric state and prevent both toxic oligomer formation and rapid fibril elongation events. Therefore, combining this approach with cell viability assays provides an ideal platform for screening therapies that can inhibit medin aggregation and stabilize medin in a non-toxic state by inhibiting specific microscopic steps. Additionally, the kinetic approach can be used to study how interactions with heterogeneous components of the extracellular milieu modulate medin aggregation, including Aβ isoforms and extracellular chaperones, and therefore shed light on medin amyloid formation in a more physiological context.

In the context of the extracellular environment, salt titration experiments demonstrate the possibility that amyloid fibril formation in medin, which has no net charge at physiological pH (pI = 6.16, with four positively charged and four negatively charged amino acid side chains in the primary structure), is driven mostly by hydrophobic interactions. This is also corroborated by the absence of any charged residues and the abundance of hydrophobic and aromatic residues such as Val, Ile, and Phe, Trp, respectively, in the amyloid-prone regions of medin as reported previously in literature through computational predictions and biophysical characterization[43].

In summary, in this study, we presented a quantitative kinetic dissection of medin amyloid formation, mechanistically justifying the prolific deposition of medin in the aging vasculature. By extending the chemical kinetics framework to an understudied but biologically important peptide, we demonstrated how chemical kinetics can be used to uncover quantitative mechanisms in protein self-assembly. We anticipate that our microscopic analysis of medin aggregation will enable a robust platform to investigate cross-seeding interactions with other amyloids, evaluate environmental

modulators of aggregation, and design microscopic step-specific pharmacological chaperones, thus contributing broadly to the fields of chemical biology and supramolecular chemistry.

## Methods
### Recombinant production of medin
The gene sequence coding for NT FlSp domain from *Nephila clavipes* with the D40K and K65D mutations (NT*$_{FlSp}$) was obtained by sequencing a pT7 plasmid encoding this gene[52], which was a kind gift from Dr. Henrik Biverstål (Karolinska Institutet, Sweden). A 606 bp DNA construct encoding a fusion protein containing, from N- to C-terminus, His$_6$-tag, NT*$_{Flsp}$, TEV protease recognition site, and human medin, was cloned into a pET-28a(+) plasmid between *Nco*I restriction site at the 5′ end and *Xho*I site at the 3′ end by GenScript Biotech, United Kingdom (plasmid deposition to Addgene underway; ID 251408). Protein production and purification were carried out as described previously for Aβ isoforms[52]. Briefly, the plasmid was transformed into *E. coli* BL21(DE3) cells (New England Biolabs, United Kingdom). Overnight-grown liquid cultures of the transformed cells were inoculated at 1:100 dilution in Luria Bertani (LB) medium containing 70 μg/mL kanamycin and grown at 30 °C and 120 rpm (Infors HT, United Kingdom) until the optical density at 600 nm (OD$_{600\ nm}$) reached 0.8–0.9. The cultures were cooled down to 20 °C and gene expression was induced with 0.1 mM isopropyl β-D-1-thiogalactopyranoside (IPTG), followed by overnight incubation. Cells were isolated by centrifugation at 5000 × *g* for 20 min at 4 °C, resuspended in 20 mL of 20 mM Tris-HCl pH 8.0 per litre of culture, re-centrifuged at ~4700 × *g* for 20 min at 4 °C, and stored at −20 °C. To check for protein expression, an aliquot of cells was collected before and after induction, centrifuged, washed, and re-centrifuged, followed by sodium dodecyl sulfate polyacrylamide gel electrophoresis (SDS-PAGE) analysis of the final pellets to detect the presence of a band corresponding to the fusion protein only in the induced cells.

The frozen cells were thawed on ice, resuspended in 40 mL of 20 mM Tris-HCl 8 M urea pH 8.0 per litre of culture supplemented with EDTA-free protease inhibitors (Roche, Germany), and sonicated to obtain a clear lysate. Precipitated materials were sedimented by centrifugation at 53,343 × *g* for 15 min at 4 °C. The supernatant was filtered through a 0.45 μm PES membrane under vacuum and loaded onto two 5 mL HisTrap Excel columns (Cytiva, Sweden) connected in series and pre-equilibrated with 20 mM Tris-HCl 8 M urea 15 mM imidazole pH 8.0. For 4 L or more culture volumes, four such columns were daisy-chained. The fusion protein was eluted with 300 mM imidazole buffer, dialyzed into 20 mM Tris-HCl pH 8.0, and cleaved with TEV protease (GenScript Biotech, United Kingdom). The reaction mixture was lyophilized and re-dissolved in 6 M guanidium chloride (GdmCl) before loading onto a Superdex 75 pg 16/600 size-

exclusion column (Cytiva, Sweden) at 1 mL/min flow rate. Monomeric medin peptide was eluted in 20 mM sodium phosphate (NaPi) 0.2 mM EDTA pH 8.0, flash-frozen in liquid nitrogen, lyophilized, and stored at −80 °C until further use. Peptide purity was confirmed by SDS-PAGE. Peptide integrity was confirmed by liquid chromatography-mass spectrometry (LC-MS). Peptide sequencing was performed on gel-eluted protein bands by the Cambridge Centre for Proteomics, Department of Biochemistry, University of Cambridge.

## Size exclusion chromatography (SEC)
Lyophilized medin was resuspended in 6 M GdmCl and subjected to size exclusion chromatography using a Superdex™ 75 Increase 10/300 GL column (Cytiva, Sweden) at 0.5 mL/min flow rate. Purely monomeric medin was eluted in aggregation assay buffer, i.e., 20 mM NaPi 0.2 mM EDTA pH 7.4, at a retention volume of 14.5 mL (Supplementary Fig. 1A).

## Dynamic light scattering (DLS)
10 µM medin monomer was either directly subjected to DLS or incubated in a 96-well plate with a non-binding surface (Corning 3881, United Kingdom) at 100 µL per well at 37 °C in quiescent conditions. Samples were withdrawn at different time intervals, and the particle size distribution was monitored at 20 °C in a Malvern Zetasizer Nano instrument (Malvern Instruments Limited, United Kingdom) at 70 µL volume ($N = 3$, $n = 3$). Two outliers were removed manually before data analysis.

## Circular dichroism (CD) spectroscopy
Far-UV CD spectra of each of 10 µM medin monomer and sonicated fibrils (grown in the absence of ThT; see "Aggregation assays and preparation of seeds" for preparation of sonicated fibrils) were acquired in a Chirascan spectrophotometer (Applied Photophysics, United Kingdom) over the wavelength range of 190–250 nm and in seven repeats, which were averaged and blank-corrected. Mean residue molar ellipticity (MRME) was calculated in $deg\ cm^2\ dmol^{-1}\ res^{-1}$ using the following formula:

$$MRME = \frac{\theta \times M}{n \times 10 \times l \times c} \quad (1)$$

where $\theta$ is the ellipticity in $mdeg$, $M$ is the average molecular weight of monomer in $g\ mol^{-1}$, $n$ is the number of residues of a monomer in $res$ unit, $l$ is the path length of the cell in $cm$, and $c$ the monomer-equivalent concentration in $g\ L^{-1}$.

## Transmission electron microscopy (TEM)
2.5 µL of 10 µM overnight-grown medin fibrils aggregated in the absence of ThT was spotted and dried for 40 s on a lacey carbon film-coated 3 mm 300-mesh copper grid, followed by washing with 2.5 µL of ultrapure water for 40 s, and then stained with 2.5 µL 2% (w/v) uranyl acetate for 40 s. Micrographs were acquired using an FEI Talos F200X G2 TEM instrument (Thermo Fisher Scientific, United Kingdom). Fibril diameter was calculated by averaging the diameters of 100 fibrils on ImageJ[77] using the TIA Reader plugin.

## Aggregation assays and preparation of seeds
All assays were carried out at 37 °C in quiescent conditions at 100 µL sample volume per well in 96-well half-area non-binding plates (Corning 3881), with an aluminum film coating to minimize evaporation. Medin monomer was incubated at different concentrations with or without pre-formed fibrils (seeds), in the presence of 20 µM ThT, with five replicates per sample. ThT fluorescence emission at 480 nm, when excited at 440 nm, was monitored using a FLUOstar Omega plate-reader (BMG Labtech GmbH, Germany) using a cycle time of 180 s with 40 flashes per reading and 0.2 s settling time per cycle. The unseeded reactions were repeated thrice using three different batches of protein.

For the preparation of seeds, 10 µM of medin monomer was incubated overnight at 37 °C in quiescent conditions without ThT at 100 µL per well in

the same plates. Amyloid aggregation was monitored using 20 µM ThT in replicate wells. The overnight-grown fibrils were sonicated for 15 s at 50% duty cycle and 10% power using a microtip sonicator (Bandelin Sonopuls, Germany) to produce seeds for subsequent seeded aggregation assays.

For salt titration experiments, 20 mM NaPi 1 M NaCl 0.2 mM EDTA pH 7.4 was diluted to the final reaction mixtures at final NaCl concentrations of 0, 40, 150, and 500 mM before incubation in replicate wells of a non-binding plate and monitoring of aggregation kinetics as described above.

## Kinetic analysis
After removing outliers manually, the resultant data from kinetic assays were normalized and analyzed using the online platform AmyloFit 2.0[46] made for global aggregation kinetic data-fitting. All data representations have been produced in GraphPad Prism 2.0. Half-times of unseeded aggregation assays were calculated in the AmyloFit 2.0 platform. For comparison of aggregation half-times of unseeded and low-seeded reactions, data were normalized and fitted using GraphPad Prism 2.0 with

$$y = y_{\min} + \frac{y_{\max} - y_{\min}}{1 + \left(\frac{t_{1/2}}{t}\right)^c} \quad (2)$$

where $y$ is the normalized ThT fluorescence and $c$ is a constant. All data were fitted with the corresponding model using 50 basin hops to ensure convergence, and the fit with the least MSE was considered the best fit. Two models were used with the consideration of the presence or absence of seeds: secondary nucleation dominated[11] and fragmentation dominated mechanisms[45]. The depolymerization rate constant $k_{off}$ was set constant to a negligible minimum of $2.8 \times 10^{-19}\,s^{-1}$ while fitting data to fragmentation-based models[46]. The average fibril length for pre-formed aggregate seeds was assumed to be 10,000 monomers[46]. High-seeded aggregation assays, where the seed:total protein ratio ranged between 23 and 33%, showed a biphasic behavior upon prolonged incubation, which may be attributed to higher-order fibril maturation events that do not form a part of the in vitro aggregation kinetic analysis (Supplementary Fig. 6). Hence, the elongation rate constant $k_+$ was calculated from the first phase.

To determine the critical concentration of medin aggregation, we first determined the ThT amplitude of sigmoidal curves of spontaneous aggregation of medin at 0 (blank), 5, 10, 15, 20, 30, 40, 75, 100, 250, 375, 500 and 625 nM in the presence of 20 µM ThT using the same conditions as described in the subsection "Aggregation assays and preparation of seeds." The ThT fluorescence intensity of the blank profile was subtracted from the rest before data fitting. Since no sigmoidal curve could be fit for the kinetic profiles at 0, 5, and 10 nM medin, the ThT amplitude in these cases were considered 0 AU (arbitrary unit). The ThT amplitude keeps increasing from 15 nM onwards; however, a two-state sigmoidal response curve similar to equation (2) could be fit considering concentrations 0–40 nM. The half-maximal point of this fit was considered to be the critical concentration of aggregation for pure monomeric medin under the given experimental conditions.

## Aβ₄₂ aggregation assays
Human $A\beta_{42}$ was recombinantly produced and purified as described previously[52] followed by aliquoting, lyophilization, and storage at −80 °C. An aliquot of lyophilized $A\beta_{42}$ was resuspended in 6 M GdmCl on ice and subjected to size exclusion chromatography on a Superdex™ 75 Increase 10/300 GL column (Cytiva, Sweden) at 0.5 mL/min flow rate. Pure monomeric $A\beta_{42}$ was eluted in aggregation assay buffer. It was incubated at different concentrations in the presence of 20 µM ThT quiescently at 37 °C in 96-well half-area non-binding plates (Corning 3881), with 100 µL per well and five replicate wells per concentration. An aluminum film coating minimized evaporation. The fluorescence emission of ThT at 480 nm upon excitation at 440 nm was monitored using a FLUOstar Omega plate-reader (BMG Labtech GmbH, Germany) using a cycle time of 300 s with 40 flashes per reading and 0.2 s settling time per cycle. Kinetic data were normalized and analyzed using GraphPad Prism 2.0 as described above to determine the

scaling exponent $\gamma$, which was found to be equal to $-1.2$ as expected[11] (Supplementary Fig. 8B).

## Annotation convention

$N$ denotes biological replicates (measurements taken from distinct samples in distinct experiments with the same set of variables). $n$ denotes technical replicates (measurements taken from the same sample split into different wells of the same plate, 100 μL per well, in the same experiment). SEM stands for standard error of mean. SD stands for standard deviation.

## Reporting summary

Further information on research design is available in the Nature Portfolio Reporting Summary linked to this article.

## Data availability

All data pertaining to the manuscript (Main and Supplementary Information) have been provided in the Supplementary Data file.

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

## Acknowledgements

V.R.C. acknowledges Bio2Brain MSC-ITN (EU Horizon 2020 Grant Agreement no. 956977) for funding this research. The authors acknowledge the EPSRC Underpinning Multi-User Equipment Call (EP/P030467/1) for funding of the Talos F200X G2 TEM instrument in the electron microscopy facility at the Yusuf Hamied Department of Chemistry, University of Cambridge. The authors would like to thank Dr. Rebecca C. Gregory and Carola Grondona for support in the Aβ42 production and Dr. Heather Greer for training on the TEM. Parts of the figures were created with BioRender.com.

## Author contributions

V.R.C. performed protein purification with support from M.P.C. V.R.C., R.I.H., and Z.T. performed aggregation kinetics assays and data analysis. V.R.C., R.I.H., S.L., and M.V. designed the project. V.R.C. and M.V. wrote the manuscript with input from all the coauthors.

## Competing interests

The authors declare no competing interests.
