## [Transparent Peer Review file · Communications Chemistry]

Rapid elongation drives the exceptionally fast aggregation of medin, the most common localized human amyloid

Corresponding Author: Professor Michele Vendruscolo

Version 0:

Reviewer comments:

Reviewer #1

(Remarks to the Author)

The manuscript describes a method of analysing aggregation data developed and established by the research group that has been published for a range of other aggregation systems but applied here to medin for the first time. The paper is therefore novel and of interest and should be published. I just have a few minor suggestion points:

1. In the first paragraph the authors mention 32 disease-related amyloid proteins, in the most recent amyloid nomenclature paper the value is now 42.
2. Could the authors expand on their statement if the ThT concentration is tailored appropriately.
3. The work was done in the absence of salt e.g. NaCl can the authors comment on how the physiological presence of salt may affect this aggregation.
4. Excitation wavelength isn't given for ThT measurements.
5. Can the authors give details about sonication conditions used rather than just brief sonication.
6. Experimental methods for data on other proteins provided in Figure 6 are not provided in the methods.

Minor grammar issue in the section: Determination of microscopic rate constants and reaction orders via kinetic modelling and global data analysis - These fibrils can in turn can catalyze

Reviewer #2

(Remarks to the Author)

The manuscript by Chowdhury et al. provides an in vitro kinetic analysis of the amyloid formation of medin, a 50 aa peptide, which forms vascular amyloid deposits in a large fraction of elderly patients. Recently, it was found to co-deposit with Amyloid-beta peptides in Alzheimer patients, which means it could be a therapeutic target in AD.

The authors developed a method for the production of recombinant medin at high purity, which is crucial for kinetic analysis. The manuscript then finds that fibril formation of medin, like Abeta and alpha-synuclein, is dominated by secondary processes, either fibril fragmentation or saturated secondary nucleation. However, the kinetic analysis alone could not distinguish between these possibilities. The paper presents a solid set of data to characterize medin amyloid formation and I did not pick up any methodological flaws.

However, I was a bit disappointed by the limited scope of the manuscript. The authors did not attempt to supplement the bulk kinetic data with real-time fluorescence or super-resolution microscopy to resolve the question whether medin aggregates by secondary nucleation or fibril fragmentation, nor did they analyse a possible change in reaction order in response to a change in agitation, which could provided further clues towards the mechanism. Many of these techniques were pioneered at Cambridge, so their omission is quite striking.

Recently, it was discovered that medin and Abeta co-deposit in vivo, raising the possibility of cross-seeding between both peptides. The authors state that the current manuscript will provide the starting point for future cross-seeding experiments, but, similar to my previous comment, I would have really like to see these data here.

The authors find that medin forms amyloid at unusually low nM concentrations, suggesting a very low critical concentration. However, an experiment to determine the critical concentration is missing.

In their discussion, the authors claim that their mechanistic analysis provides the mechanism for the formation of toxic medin oligomers. In my view, this argument needs to be refined. The data shows that medin has an uncommonly low critical concentration and a low nucleation barrier. As the authors correctly conclude, this would mean very few free oligomers should exist in solution. So, far from providing an obvious mechanism for the formation of toxic oligomers, to me the data actually make it more mysterious how and why medin could be neurotoxic. This is an intriguing point worth discussing in more detail.

Minor point:

Since this is not a review article, figure 1 feels unnecessary.

Reviewer #3

(Remarks to the Author)

The authors present an interesting study on the mechanism of medin amyloid fibril formation. I have a few comments and questions regarding the manuscript, which should be taken into consideration before publication, especially regarding the monomeric/oligomeric state of the protein and the aggregation kinetics.

Major points

1. The authors state that the initial solution of medin contained both monomeric, as well as various oligomeric forms of the protein, which is also seen in the DLS analysis. Did the authors determine what was the ratio of monomers versus oligomers in the initial reaction solutions (by size-exclusion or native PAGE)? If the part of oligomers is not negligible, is it correct to refer to the initial protein solutions as "monomers" in both the figures and text?
2. If medin exists in monomeric and oligomeric forms, while only the monomers become incorporated into aggregates via nucleation, secondary nucleation and elongation, what role do the oligomers play? Are they quasi-stable and become monomers when the equilibrium shifts? This is important, as the fitting procedure, as well as the obtained reaction constants rely on knowing the exact concentration of available monomers. If the oligomer part of the reaction solution does not participate in the reaction, then the actual monomer concentration is lower than the stated total protein concentration.
3. Could the biphasic kinetics be explained by the presence of quasi-stable oligomers? The highest concentration samples would normally have higher oligomer concentrations, which fits with the presence of the biphasic data. Is it possible that all available monomers quickly become incorporated into fibrils, which then causes the sudden shift in monomer-oligomer equilibrium to produce more monomers to be incorporated into fibrils? In this case, the second part of the biphasic kinetics would not correspond to fibril maturation and the end of the first part could not be regarded as the incorporation of all available non-aggregated protein.
4. In Figure 4, the authors show heavily seeded aggregation kinetics. I have two concerns regarding this part. First, the normalized kinetics appear to almost overlap with each other and it's very difficult to see if the fit correlates with the data. It is therefore not certain that the determination of the constant is correct. The second question is about the actual kinetic monitoring. If medin aggregates very rapidly even on its own, are the kinetic monitoring procedures accurate when the reaction is even more rapid with the addition of high seed concentrations? Is it possible that aggregation begins during the sample preparation, distribution, instrument set-up procedure and time required for thermal equilibrium, which skews the obtained results?
5. In Figure 4B, both the Secondary-nucleation dominated, as well as Fragmentation dominated fits are very accurate for the middle concentrations and diverge quite heavily towards the higher and lower protein concentrations. The 50 nM and 625 nM kinetic curves are far from the fit curves by their lag times, T1/2 positions and slopes. I assume the authors have attempted to change the models and parameters to achieve the best fit. Is this result the best possible model/parameter variant or can it be refined further by changing other settings in AmyloFit? It would be interesting for the reader to include a bit more information regarding the attempted models and refinements.

Minor points

1. The Supporting data document has Figure S5 shown twice.
2. Introduction line "...and sustained by saturated secondary processes that proceed independently of monomer concentration (reaction order ~0)." I assume this statement applies to the tested range of concentrations and the process is not independent of monomer concentration under all possible concentrations. Would the process still be saturated at very low concentrations of the non-aggregated protein?

Version 1:

Reviewer comments:

Reviewer #1

(Remarks to the Author)

I am happy with the authors responses and modifications made.

Reviewer #2

(Remarks to the Author)

In their revised manuscript, the authors have added data and additional analysis to address the most important points in my critique, which substantially improved the manuscript. I support the acceptance of the revised manuscript for publication.

Reviewer #3

(Remarks to the Author)

The authors have significantly improved the manuscript in areas that were previously lacking clarity. The manuscript is suitable for publication in its current state.

Reviewer #1

The manuscript describes a method of analysing aggregation data developed and established by the research group that has been published for a range of other aggregation systems but applied here to medin for the first time. The paper is therefore novel and of interest and should be published.

We thank the reviewer for the positive opinion about our manuscript.

I just have a few minor suggestion points:

1. In the first paragraph the authors mention 32 disease-related amyloid proteins, in the most recent amyloid nomenclature paper the value is now 42.

We thank the reviewer for pointing this out. We have now corrected this in the manuscript and have cited Buxbaum, JN *et al.*, *Amyloid*, 2024.

2. Could the authors expand on their statement if the ThT concentration is tailored appropriately.

We have chosen 20 μM as the standard ThT concentration for all measurements reported in this manuscript. We chose 20 μM ThT as this is greater than the medin concentration used for kinetics assays ($<1 \mu\text{M}$) and for growing seed fibrils (10 μM), so that medin and not ThT is the limiting reagent. At 20 μM , fluorescence emission of ThT at 480 nm increases linearly with medin fibril concentration as shown in Figure S1.

3. The work was done in the absence of salt e.g. NaCl can the authors comment on how the physiological presence of salt may affect this aggregation.

We have now included information about this in Results (subsection “Saturated secondary pathways sustain medin aggregation *in vitro*”), Discussion, Methods (subsection “Aggregation assays and preparation of seeds”) and Supplementary Information (Figure S5 and Tables S2 and S3).

4. Excitation wavelength isn't given for ThT measurements.

We thank the reviewer for pointing this out. We have now clarified the excitation wavelength in the Methods (subsections “Aggregation assays and preparation of seeds” and “A β_{42} aggregation assays”).

5. Can the authors give details about sonication conditions used rather than just brief sonication.

We have now mentioned the sonication conditions in the Methods (subsection “Aggregation assays and preparation of seeds”).

6. Experimental methods for data on other proteins provided in Figure 6 are not provided in the methods.

We have now mentioned in the caption for Figure 6A that these values are obtained from literature. For clarification, we also mention the papers here:

A β ₄₂: Cohen, SIA *et al.*, *PNAS*, 2013, A β ₄₀: Meisl, G *et al.*, *PNAS*, 2011, Tau: Kundel, F *et al.*, *ACS Chem Neurosci*, 2018, α Syn: Buell, AK *et al.*, *PNAS*, 2014

Minor grammar issue in the section: Determination of microscopic rate constants and reaction orders via kinetic modelling and global data analysis - These fibrils can in turn can catalyze
We thank the reviewer for pointing out this flaw. We have now corrected this in the manuscript.

Reviewer #2

The manuscript by Chowdhury *et al.* provides an *in vitro* kinetic analysis of the amyloid formation of medin, a 50 aa peptide, which forms vascular amyloid deposits in a large fraction of elderly patients. Recently, it was found to co-deposit with Amyloid-beta peptides in Alzheimer patients, which means it could be a therapeutic target in AD.

The authors developed a method for the production of recombinant medin at high purity, which is crucial for kinetic analysis. The manuscript then finds that fibril formation of medin, like A β and alpha-synuclein, is dominated by secondary processes, either fibril fragmentation or saturated secondary nucleation. However, the kinetic analysis alone could not distinguish between these possibilities. The paper presents a solid set of data to characterize medin amyloid formation and I did not pick up any methodological flaws.

We thank the reviewer for this positive opinion about our manuscript.

We note that primary processes ($k_+k_n m_0^{n_c} = 0.5 \times m_0 s^{-2}$) and secondary processes ($k_+k_2 m_0^{n_2+1} = 2 \times m_0 s^{-2}$) promote aggregation to the similar levels. This is unlike A β ₄₂ where aggregation is dominated by secondary nucleation (rate of primary processes $k_+k_n m_0^{n_c} = 900 \times m_0^2 M^{-1} s^{-2}$ vs rate of secondary processes $k_+k_2 m_0^{n_2+1} = 3 \times 10^{10} \times m_0^2 M^{-1} s^{-2}$) as secondary processes are 10⁷ orders faster than primary processes (Meisl, G *et al.*, *PNAS*, 2014). This is why we describe how secondary pathways sustain aggregation (in addition to primary processes) rather than dominating it. We hope that this clarification is helpful.

However, I was a bit disappointed by the limited scope of the manuscript. The authors did not attempt to supplement the bulk kinetic data with real-time fluorescence or super-resolution microscopy to resolve the question whether medin aggregates by secondary nucleation or fibril fragmentation, nor did they analyse a possible change in reaction order in response to a change in agitation, which could provide further clues towards the mechanism. Many of these techniques were pioneered at Cambridge, so their omission is quite striking.

We appreciate the reviewer's suggestions. The main conclusion of our study is that the fast fibril growth of medin is dominated by elongation, as quantified directly from high-seeded assays where elongation is the main mechanism. This elongation rate exceeds those reported for other pathological amyloids and accounts for the rapid aggregation we observe at nanomolar concentrations.

Within our measured concentration range (25-625 nM), secondary pathways are saturated and weakly monomer-dependent (reaction order near 0), and global fits show that fragmentation-dominated and secondary-nucleation-dominated models describe the data equally well (similar MSEs). Consequently, discriminating which saturated secondary route is more active is not critical for our claims, which rest on elongation dominance and the quantitative rate parameters we report. We now explain this point more explicitly in the Discussion, and we frame methodologically richer discrimination between saturated secondary mechanisms as valuable future work rather than a requirement for the present conclusions.

Recently, it was discovered that medin and Abeta co-deposit *in vivo*, raising the possibility of cross-seeding between both peptides. The authors state that the current manuscript will provide the starting point for future cross-seeding experiments, but, similar to my previous comment, I would have really like to see these data here.

We appreciate the reviewer's suggestion. In Figure 5 we have broken down the amyloid formation pathway of medin into multiple intermediate steps (monomers, primary and secondary oligomers, protofilaments and mature amyloid fibrils). Our proposed mechanism of medin aggregation suggests two possible routes of cross-seeding: (i) Exposed nucleation sites on monomers and small soluble oligomers seed heterogeneous nucleation of A β , serum amyloid A or transthyretin, which are known co-aggregators of medin (Wagner, J *et al.*, *Nature*, 2022; Larsson, A *et al.*, *Amyloid*, 2011; Maerivoet, A *et al.*, *Amyloid*, 2025), and (ii) The fibrils of medin form novel surfaces for the co-aggregating protein to nucleate.

Depending on which intermediate is more efficient at cross-seeding, the therapeutic route to combat the co-aggregation will differ. For example, if the monomers are involved in cross-seeding, then we would need to use pharmacological chaperones that either stabilise the monomers in a non-toxic state with the nucleation sites masked, or that promote fibril formation followed by clearance *in vivo*. On the other hand, if the fibrils are the most effective surfaces for heterogeneous nucleation, then the therapeutic compounds should either prevent fibril formation or act as site-specific blockers of heterogeneous nucleation. All these points form a framework for future studies based on insights from Figure 5 of this manuscript.

The authors find that medin forms amyloid at unusually low nM concentrations, suggesting a very low critical concentration. However, an experiment to determine the critical concentration is missing.

We have now added information about the critical medin concentration for a positive ThT amplitude of sigmoidal aggregation curve in Figure S3D and the Methods section (subsection: "Kinetic analysis").

In their discussion, the authors claim that their mechanistic analysis provides the mechanism for the formation of toxic medin oligomers. In my view, this argument needs to be refined. The data shows that medin has an uncommonly low critical concentration and a low nucleation barrier. As the authors correctly conclude, this would mean very few free oligomers should exist in solution. So, far from providing an obvious mechanism for the formation of toxic

oligomers, to me the data actually make it more mysterious how and why medin could be neurotoxic. This is an intriguing point worth discussing in more detail.

We thank the reviewer for the insightful comment. It has been reported for A β ₄₂ aggregation that the total concentration of all oligomers is less than 1.5% of the initial monomer load, with the maximum appearing when the monomer and aggregate concentrations are equal, i.e., at $t_{1/2}$ (e.g. Cohen, SIA *et al.*, *PNAS*, 2013; Arosio, P *et al.*, *Phys Chem Chem Phys*, 2015). Previous literature has also reported that medin mediates maximal toxicity in its oligomeric state, specifically when studied on cultured human aortic smooth muscle cells (Peng, S *et al.*, *Lab Invest*, 2007, Figure 6) and bovine endothelial cells (Madine, J *et al.*, *Eur Biophys J*, 2010, Figure 4). The current manuscript, in particular Figure 5, describes both primary and secondary processes are sources of medin oligomers and they operate at rates that are of the same order of magnitude. This is unlike A β ₄₂ where the soluble neurotoxic oligomers are formed predominantly by secondary nucleation (Cohen, SIA *et al.*, *PNAS*, 2013). Overall, although the total oligomer population is expected to be small, even transient, low-abundance oligomers may be sufficient to drive toxicity if they are highly potent and interact preferentially with cellular membranes or vascular surfaces. In our mechanism, both primary and secondary pathways continuously generate such oligomeric species, providing a natural route to reconcile low steady-state oligomer concentrations with pronounced biological effects.

Minor point:

Since this is not a review article, figure 1 feels unnecessary.

We appreciate this observation, but we would prefer to retain Figure 1, since it provides a framework for understanding the relevance of medin for the readers less familiar with this protein.

Reviewer #3

The authors present an interesting study on the mechanism of medin amyloid fibril formation. I have a few comments and questions regarding the manuscript, which should be taken into consideration before publication, especially regarding the monomeric/oligomeric state of the protein and the aggregation kinetics.

Major points

1. The authors state that the initial solution of medin contained both monomeric, as well as various oligomeric forms of the protein, which is also seen in the DLS analysis. Did the authors determine what was the ratio of monomers versus oligomers in the initial reaction solutions (by size-exclusion or native PAGE)? If the part of oligomers is not negligible, is it correct to refer to the initial protein solutions as “monomers” in both the figures and text?

We thank the reviewer for asking this question. We have now added Figure S2 to provide additional evidence that medin is monomeric at the start of the aggregation reaction. Figure S2 reports a representative chromatogram of the gel-filtration experiment that we conducted on

denatured medin to isolate pure monomers right before diluting them to sub-micromolar concentrations in the reaction mixtures and monitoring ThT fluorescence, and a representative DLS profile of freshly eluted 10 μM pure monomer of medin. In practical terms, we always ensure that monitoring of ThT fluorescence is initiated within 1 hour since the elution of monomeric medin from the gel filtration column. Only when 10 μM medin monomer (concentration 2-3 orders of magnitude higher than those used in kinetic calculation experiments) is incubated for at least 1.5 hours on ice, we start noticing the first traces of higher molecular weight species as evident in the DLS experiment in Figure 2C which we refer to as oligomers.

2. If medin exists in monomeric and oligomeric forms, while only the monomers become incorporated into aggregates via nucleation, secondary nucleation and elongation, what role do the oligomers play? Are they quasi-stable and become monomers when the equilibrium shifts? This is important, as the fitting procedure, as well as the obtained reaction constants rely on knowing the exact concentration of available monomers. If the oligomer part of the reaction solution does not participate in the reaction, then the actual monomer concentration is lower than the stated total protein concentration.

We thank the reviewer for this comment. As detailed in our response to the point above, and in the new Figure S2, under the conditions used for the kinetic experiments the starting solutions are essentially purely monomeric. Medin is first denatured and then passed through size-exclusion chromatography; we collect only the centre of the monomer peak and start the aggregation assays within 1 hour of this elution. Dynamic light scattering of freshly eluted medin shows a single species with a hydrodynamic diameter consistent with monomer, and higher-molecular-weight species only appear after much longer incubations and at 10 μM , i.e. at concentrations and timescales different from those used in the kinetic measurements. Thus, within the sensitivity of SEC and DLS, there is no detectable oligomer pool present at the start of the reactions, and the total protein concentration used in the fits accurately reflects the available monomer concentration.

3. Could the biphasic kinetics be explained by the presence of quasi-stable oligomers? The highest concentration samples would normally have higher oligomer concentrations, which fits with the presence of the biphasic data. Is it possible that all available monomers quickly become incorporated into fibrils, which then causes the sudden shift in monomer-oligomer equilibrium to produce more monomers to be incorporated into fibrils? In this case, the second part of the biphasic kinetics would not correspond to fibril maturation and the end of the first part could not be regarded as the incorporation of all available non-aggregated protein.

We thank the reviewer for this thoughtful suggestion. We have considered the possibility that the biphasic behaviour in the high-seeded reactions could arise from a monomer-oligomer equilibrium. However, our experiments indicate that this explanation is not likely under the conditions used here. As shown in Fig. 2 and Fig. S2, freshly prepared medin is essentially monomeric at the start of the reaction, and no detectable oligomeric population is present in the concentration range used for the kinetic assays. We therefore do not expect significant redistribution between monomers and oligomers during the early stages of the reaction. By contrast, biphasic kinetics in highly seeded reactions are well documented for fibril-forming

systems and are typically attributed to higher-order structural maturation events (for example, changes in fibril packing or bundling) that alter the ThT signal without affecting the underlying mass of fibrils. This interpretation is consistent with our data: the reaction half-times scale as expected for elongation-dominated kinetics, and the early exponential phase used to extract the elongation rate constant is unaffected by the later ThT increase. We have added a brief note in the manuscript to clarify this point.

4. In Figure 4, the authors show heavily seeded aggregation kinetics. I have two concerns regarding this part. First, the normalized kinetics appear to almost overlap with each other and it's very difficult to see if the Fit correlates with the data. It is therefore not certain that the determination of the constant is correct. The second question is about the actual kinetic monitoring. If medin aggregates very rapidly even on its own, are the kinetic monitoring procedures accurate when the reaction is even more rapid with the addition of high seed concentrations? Is it possible that aggregation begins during the sample preparation, distribution, instrument set-up procedure and time required for thermal equilibrium, which skews the obtained results?

We thank the reviewer for making this important point. The apparent overlap of the 100, 250 and 500 nM traces in Fig. 4A reflects the fact that these reactions are elongation-dominated and therefore display very similar exponential rises, as expected for this regime. To make this point clearer, we now include a zoom of the early time points in Fig. S7, where the data and fits can be distinguished and the exponential behaviour is evident. The time between mixing and the first recorded point is on the order of tens of seconds, which corresponds to only a small fraction of the exponential rise for the seed concentrations used. We verified that shifting the time axis by this dead time has a negligible effect on the fitted elongation rate constant and does not alter our conclusions. This is further supported by the very low mean-squared errors of the fits and the excellent reproducibility of the elongation rate across independent protein preparations (Table S6).

5. In Figure 4B, both the Secondary-nucleation dominated, as well as Fragmentation dominated fits are very accurate for the middle concentrations and diverge quite heavily towards the higher and lower protein concentrations. The 50 nM and 625 nM kinetic curves are far from the fit curves by their lag times, $T_{1/2}$ positions and slopes. I assume the authors have attempted to change the models and parameters to achieve the best fit. Is this result the best possible model/parameter variant or can it be refined further by changing other settings in AmyloFit? It would be interesting for the reader to include a bit more information regarding the attempted models and refinements.

We confirm that we explored different models and parameter sets within AmyloFit and this is the best global fit we can obtain, using 50 basin hops to ensure convergence and minimise the MSE. We agree that the model deviates from the data at the lowest (50 nM) and highest (625 nM) concentrations. This likely reflects the limits of the simple global model at the edges of the kinetic regime, where signal-to-noise is lowest (50 nM) and where additional higher-order processes may become more prominent (625 nM). We now note in the Results that our quantitative conclusions about rate constants and reaction orders are primarily constrained by the well-fitted intermediate concentrations, and are not driven by the boundary points.

Minor points

1. The Supporting data document has Figure S5 shown twice.

We apologise for this mistake. We have now corrected it.

2. Introduction line "...and sustained by saturated secondary processes that proceed independently of monomer concentration (reaction order ~ 0)." I assume this statement applies to the tested range of concentrations and the process is not independent of monomer concentration under all possible concentrations. Would the process still be saturated at very low concentrations of the non-aggregated protein?

We now estimate the critical concentration of medin monomer required to obtain a positive ThT fluorescence to be about 14 nM (Figure S3D). Our kinetic regime was defined to be 25-625 nM (Figure S3A, B), as in this concentration range, $\ln t_{1/2}$ scales negatively with $\ln m_0$ with a slope of -0.5. The reason why we did not investigate concentrations below 25 nM is because at such low concentrations, different replicates attain plateau at very different timescales, thus making the resultant data noisy and unreliable. It is therefore difficult to predict if the secondary pathways will still be saturated in the 11 nM window between the critical concentration and the lower limit of the kinetic regime.

Reviewer #1: I am happy with the authors responses and modifications made.

Reviewer #2: In their revised manuscript, the authors have added data and additional analysis to address the most important points in my critique, which substantially improved the manuscript. I support the acceptance of the revised manuscript for publication.

Reviewer #3: The authors have significantly improved the manuscript in areas that were previously lacking clarity. The manuscript is suitable for publication in its current state.

We are grateful to the three referees for recommending publication of our manuscript.